# Combined Transcriptomic and Protein Array Cytokine Profiling of Human Stem Cells from Dental Apical Papilla Modulated by Oral Bacteria

**DOI:** 10.3390/ijms23095098

**Published:** 2022-05-03

**Authors:** Valeriia Zymovets, Yelyzaveta Razghonova, Olena Rakhimova, Karthik Aripaka, Lokeshwaran Manoharan, Peyman Kelk, Maréne Landström, Nelly Romani Vestman

**Affiliations:** 1Department of Odontology, Umeå University, 901 85 Umea, Sweden; valeriia.zymovets@umu.se (V.Z.); olena.rakhimova@umu.se (O.R.); 2Department of Microbiology, Virology and Biotechnology, Mechnikov National University, 65058 Odesa, Ukraine; e.razghonova@ukr.net; 3Department of Medical Biosciences, Umeå University, 901 85 Umea, Sweden; karthik.aripaka@umu.se (K.A.); marene.landstrom@umu.se (M.L.); 4National Bioinformatics Infrastructure Sweden, Lund University, 223 62 Lund, Sweden; lokeshwaran.manoharan@nbis.se; 5Section for Anatomy, Department of Integrative Medical Biology (IMB), Umea University, 901 87 Umea, Sweden; peyman.kelk@umu.se; 6Department of Endodontics, Region of Västerbotten, 901 85 Umea, Sweden; 7Wallenberg Centre for Molecular Medicine, Umeå University, 901 87 Umea, Sweden

**Keywords:** stem cells from the apical papilla (SCAP), cytokine secretion, regenerative endodontic treatment (RET), endodontics, *Fusobacterium nucleatum*, immune response, osteogenic potential, transcriptome analysis, IL-6, IL-8

## Abstract

Stem cells from the apical papilla (SCAP) are a promising resource for use in regenerative endodontic treatment (RET) that may be adversely affected by oral bacteria, which in turn can exert an effect on the success of RET. Our work aims to study the cytokine profile of SCAP upon exposure to oral bacteria and their supernatants—*Fusobacterium nucleatum* and *Enterococcus faecalis*—as well as to establish their effect on the osteogenic and immunogenic potentials of SCAP. Further, we target the presence of key proteins of the Wnt/β-Catenin, TGF-β, and NF-κB signaling pathways, which play a crucial role in adult osteogenic differentiation of mesenchymal stem cells, using the Western blot (WB) technique. The membrane-based sandwich immunoassay and transcriptomic analysis showed that, under the influence of *F. nucleatum* (both bacteria and supernatant), the production of pro-inflammatory cytokines IL-6, IL-8, and MCP-1 occurred, which was also confirmed at the mRNA level. Conversely, *E. faecalis* reduced the secretion of the aforementioned cytokines at both mRNA and protein levels. WB analysis showed that SCAP co-cultivation with *E. faecalis* led to a decrease in the level of the key proteins of the Wnt/β-Catenin and NF-κB signaling pathways: β-Catenin (*p* = 0.0068 *), LRP-5 (*p* = 0.0059 **), and LRP-6 (*p* = 0.0329 *), as well as NF-kB (*p* = 0.0034 **) and TRAF6 (*p* = 0.0285 *). These results suggest that oral bacteria can up- and downregulate the immune and inflammatory responses of SCAP, as well as influence the osteogenic potential of SCAP, which may negatively regulate the success of RET.

## 1. Introduction

The oral cavity is home to a complex microbiome that lives in a mutually beneficial state with the host [1]. However, in cases of oral tissue injury or inflammatory diseases, e.g., dental trauma or caries, the oral microbiota may invade the dental root canal, causing an endodontic infection followed by pulp necrosis and bone resorption (apical periodontitis) [2]. When pulp necrosis occurs in young patients with permanent immature teeth, further root maturation ceases, which in turn has an impact on tooth survival [3]. 

Nowadays, therapeutic biological procedures such as regenerative endodontic treatment (RET) take advantage of stem cell dynamics of the tooth’s periapical area. In this context, RET aims to eliminate root canal infection and promote root development through a cascade of well-orchestrated biological events mediated by stem cells surviving the biofilm-induced inflammatory conditions [4]. For this purpose, aside from their ability to coordinate regeneration processes, mesenchymal stem cells (MSCs) contribute by being critical players in host defense and inflammation [5]. Thus, when injury or infection occurs, stem cells come in close proximity to bacteria and bacterial components, regulating tissue regeneration even under such harsh environmental conditions [6]. 

Stem cells from the apical papilla (SCAP), MSCs isolated from teeth papilla tissue, are described as a promising tool for RET and regeneration in general. SCAP are characterized by their plasticity, potency, and versatility in superior contrast to dental pulp stem cells (DPSCs) [7]. Interestingly, MSCs’ proliferative potential and multilinear differentiation, as well as the production of proinflammatory cytokines, seem to be associated with the bacterial strains, as well as the specific bacterial components, to which they are exposed [8]. It has been described that bacterial components act on cells via the binding and activation of specific receptors, including Toll-like receptors [9], which in turn activates major inflammatory transcription factors and regulates the expression of pro-inflammatory genes such as interleukin-6 (IL6), interleukin-8 (IL8), and tumor necrosis factor alpha (TNF-α) [10].

*Fusobacterium nucleatum* is a high-frequency specie isolated from the oral cavity but usually absent or very rarely detected elsewhere in the human body [11]. *F. nucleatum* is closely related to the development of periodontitis [12] and endodontic infections [13] and is described as a possible oncogene related to colorectal cancer, pancreatic cancer, and oral cancers [14]. This Gram-negative bacterial specie is able to adhere to a variety of oral microbes and can produce a group of virulence factors and toxic metabolites, including lipopolysaccharides, porins, butyrate, and propionate ammonia [15]. Enterococcal species are core constituents of the intestinal flora and have gained notoriety as frequent causes of multiple antibiotic-resistant, hospital-acquired infections [16]. In the oral cavity, Enterococcus species, in particular *Enterococcus faecalis*, have been found to be associated with chronic apical periodontitis and failed root canal treatments [17]. 

*F. nucleatum* has been described as a dominant microorganism in periodontal tissue; however, its prevalence and relative abundance in periodontal health and periodontitis has challenged its specific role in each stage of periodontal disease [18]. Although the composition of the microbiota does not always reflect the clinical diagnosis criteria, new efforts have been made to critically understand the connection between periodontal inflammation and biofilm composition [19]. Accordingly, a new model termed “Inflammation-Mediated Polymicrobial-Emergence and Dysbiotic-Exacerbation” (IMPEDE) was recently reported and complements the 2017 World Workshop Classification of Periodontitis [20]. This model suggests that patients presenting gingivitis (stage 1) are associated with an overgrowth of commensal plaque bacteria, while patients presenting early periodontitis (stage 2) are associated with polymicrobial diversity. In later stages of the disease (stages 3 and 4), a decrease in polymicrobial diversity has been associated with an overgrowth of specific periodontal pathogens [19]. In this context, *F. nucleatum* is considered an important species for its co-aggregation capacity, being a bridge between symbiotes and true pathogens [21].

In a previous study, we reported that key species in dental root canal infection, namely *Actinomyces gerensceriae*, *Slackia exigua*, *F. nucleatum*, and *E. faecalis* were able to modulate SCAP under oxygen-free conditions in a species-dependent fashion. Moreover, *E. faecalis* and *F. nucleatum* exhibited the strongest binding capacities, resulting in significantly reduced SCAP proliferation. Notably, *F. nucleatum*, but not *E. faecalis*, induced production of the proinflammatory chemokines IL-8 and IL-10 from SCAP [22]. Accordingly, results from the analyzed cytokines should be expanded and the effects of a broad variety of inflammatory cytokines further explored.

In this study, we examined the cytokine expression profile generated by SCAP in response to clinically relevant bacterial species, namely *F. nucleatum* and *E. faecalis*, using mRNA transcriptome analysis and a cytokine antibody array. Further, we analyzed whether the secreted cytokines and chemokines had a significant role in the osteogenic and immunomodulatory potentials of SCAP. In addition, key proteins involved in signaling pathways playing an important role in the adult osteogenic differentiation of mesenchymal stem cells were analyzed by Western blot. Thus, a better understanding of cell behavior at sites of bacterial infection appears to be a key strategy for the development of new approaches for RET. Interestingly, this field remains largely unexplored. 

## 2. Results

### 2.1. Oral Bacteria Isolated from Root Canals Modulate SCAP Cytokine Profiling and Their Osteogenic and Immunomodulatory Potentials in a Species-Dependent Manner

To examine the cytokine expression profile, we used a membrane-based sandwich immunoassay (Proteome Profiler Human XL Cytokine Array). We treated the membranes with conditioned medium from SCAP co-cultivated with viable *F. nucleatum* or *E. faecalis*, as well as *F. nucleatum* supernatants (Figure 1).

The production of eight cytokines, including IL-8, IL-6, monocyte chemoattractant protein-1 (MCP-1), growth-related oncogene-alpha (GROalfa), Emmprin, granulocyte-macrophage colony-stimulating factor (GM-CSF), vascular cell adhesion molecule-1 (VCAM-1), and urokinase-type plasminogen activator receptor (uPAR), was significantly upregulated for SCAPs when co-cultured with viable *F. nucleatum*, while thrombospondin-1 was downregulated (Figure 1A). Moreover, IL-8, IL-6, MCP-1, Dickkopf-1 (Dkk-1), Emmprin, and stromal-cell-derived factor-1α (SDF-1α) were upregulated in SCAP co-cultured with supernatants of *F. nucleatum*, while interleukin-17 (IL-17) and vascular endothelial growth factor (VEGF) were downregulated (Figure 1B).

Compared to the control, *E.* faecalis upregulated the production of only three cytokines—Emmprin, migration inhibitory factor (MIF), and endoglin—for SCAP (Figure 1C) and downregulated the production of four cytokines—VEGF, angiogenin, pentraxin 3, and SDF-1α.

Moreover, to further investigate the mechanism behind the bacterial effect on SCAP, the secreted cytokines were analyzed based on their osteogenic and immunomodulatory potential (Figure 2). Among the cytokines produced by SCAP modulated by viable *F. nucleatum,* GM-CSF, VCAM-1, and uPAR had exclusively immunomodulatory potential, and Emmprin played a significant role in osteogenesis. The cytokines IL-8, IL-6, MCP-1, and GROalfa were associated with both immunomodulation and osteogenesis. Thrombospondin (TSP-1), playing a role in immunomodulation, was the only cytokine significantly downregulated compared to SCAP without bacterial stimulation (Figure 2A). 

Among cytokines upregulated by SCAP modulated by *F. nucleatum* supernatants, Dkk-1 and Emmprin were exclusively related to osteogenesis, while the other six had both osteogenic and immunomodulatory potentials. The cytokines VEGF and IL-17, which had immunomodulatory roles, were significantly downregulated (Figure 2B). Thus, detection of the pro-inflammatory cytokines IL-6 and IL-8 was exclusively involved with *F. nucleatum* infection. Interestingly, Dkk-1 doubled its concentration when cells were exposed to *F. nucleatum* supernatants compared to the control. Dkk-1 is an inhibitor of Wnt signaling, which is associated with osteoblast differentiation [23].

Only three cytokines were upregulated when SCAP were co-cultured with viable *E. faecalis.* Emmprin was osteogenesis-related, MIF was immunomodulatory-related, and endoglin played a role in both osteogenesis and immunomodulation. Five cytokines that played roles in osteogenesis (angiogenin), immunomodulation (VEGF and thrombospondin-1), and both osteogenesis and immunomodulation (pentraxin 3 and SDF-1α) were significantly downregulated in SCAP co-cultured with viable *E. faecalis* (Figure 2C). Notably, *E. faecalis* upregulated the expression of the immunosuppressive cytokines MIF and endoglin; however, neither the pro-inflammatory IL-8 nor IL-6 was secreted by SCAP co-cultured with *E. faecalis.*

### 2.2. F. nucleatum Triggers While E. faecalis Inhibits Pro-Inflammatory Chemokine and Cytokine Generation, Such as IL-6, IL-8, and MCP-1, at the mRNA and Protein Levels

Data about the mRNA expression of the corresponding 105 cytokines and chemokines examined by cytokine array were analyzed at the mRNA level by transcriptomics (Figure 3). The transcriptome analysis profile showed 23 cytokines upregulated and only 3 downregulated when SCAP were co-cultured with viable *F. nucleatum* compared to the control. Accordingly, IL-8, MIP-3a, RANTES, and IL-6 showed the highest levels of secretion in SCAP, while TfR, C5/C5a, and IL-16 were downregulated (Figure 3A). Furthermore, 10 cytokines were upregulated when SCAP were co-cultured with *F. nucleatum* supernatants, including BDNF, IL-32, GROα, ICAM-1, LIF, MCP-1, IL-6, VCAM-1, and GDF-15, and IL-8 was secreted at the highest level (Figure 3B). As expected, when SCAP were co-cultured with viable *E. faecalis*, only five cytokines were upregulated, including BDNF, IL-18 Bpa, VCAM-1, RAGE, and GDF-15, while two were downregulated (IGFBP-3 and C5/C5a) (Figure 3C).

### 2.3. SCAP Modulated by F. nucleatum Express Cytokine mRNA and Translate It into Protein, as Shown for IL-6, IL-8, GM-CSF, and MCP-1. However, the Gene and Protein Expressions Diverged When SCAP Were Modulated by E. faecalis

Assessing mRNA levels only gives information on transcribing DNA to mRNA but not on further protein production and secretion. Thus, we compared the mRNA expression for a set of 105 cytokines and chemokines with the corresponding protein expression using a cytokine array and transcriptomic data. The comparisons between cytokine mRNA and protein expression levels performed on SCAP showed diverging results (Figure 4). Intriguingly, increased mRNA levels for cytokines did not always correlate with protein production. Twenty-five cytokines and chemokines showed upregulation at the 

mRNA level, while only twelve had detectable upregulation at the protein level (Figure 5). 

GM-CSF, GROα, IL-6, IL-8, MCP-1, and VCAM-1 were upregulated when SCAP were co-cultured with viable *F. nucleatum*, as detected by microarray and transcriptomics. IL-6, IL-8, and MCP-1 showed detectable upregulation in SCAP co-cultured with *F. nucleatum* supernatants. Notably, SCAP did not express cytokine mRNA and translate it into protein when cells were co-cultured with *E. faecalis* (Figure 4).

### 2.4. Biological Processes Such as Immune Response, Inflammatory Response, and Response to Hypoxia Processes Were Decreased When SCAP Were Modulated by E. faecalis but Enhanced in Cases of SCAP Co-Cultured with F. nucleatum

The immune response, inflammatory response, and response to hypoxia processes were decreased in cases of SCAP co-cultured with *E. faecalis* but enhanced in cases of SCAP co-cultured with viable *F. nucleatum*, as interpreted by GO enrichment analysis.

We used Proteomaps, an open web resource, which enabled us to obtain a picture of the composition of pathways and cellular processes [24]. Compared to the control (Figure 6A), the proteomic analysis predicted that CD molecules, cytokine–cytokine receptor interaction, the NF-kappa B signaling pathway, the TNF signaling pathway, and messenger RNA biogenesis were among the highly expressed proteins and pathways in SCAP co-cultured with viable *F. nucleatum,* while the Rap1 signaling pathway was downregulated (Figure 6B). The co-culturing of SCAP with supernatant of *F. nucleatum* led to the upregulation of CD molecules, as well as the TNF and Wnt signaling pathways, but downregulation of the Ras signaling pathway (Figure 6C).

The TGF-β1 signaling pathway, CD molecules, and amino acid metabolism were upregulated when SCAP were co-cultured with live *E. faecalis*, while the Rap 1, NF-kappa B, and Ras signaling pathways were downregulated (Figure 6D). 

Additionally, GO enrichment analysis was used for interpreting the data and generating hypotheses underlying biological processes (Figure 7). Accordingly, all variants of the SCAP treatment led to an increase in angiogenesis, chemotaxis, and negative regulation of the apoptotic process in comparison with the non-treated control. The immune response, inflammatory response, and response to hypoxia processes were decreased in cases of SCAP co-cultured with *E. faecalis* but enhanced in cases of SCAP co-cultured with viable *F. nucleatum* or supernatant. In addition, processes of cellular response for interleukin-1, cellular response to lipopolysaccharide, and cellular response to TNF processes were only detected in SCAP in the presence of *F. nucleatum* (viable bacteria and supernatant).

### 2.5. E. faecalis Led to the Decrease of Key Protein Levels of Wnt/β-Catenin and NF-κB Signaling Pathways Playing Crucial Role in Bone Formation

Based on the downstream pathways that were predicted by the Proteomaps as upregulated or downregulated in response to the secreted cytokines, the SCAP cell lysates were probed for the presence of key proteins of the Wnt/β-Catenin signaling pathway, the TGF-β signaling pathway, and the NF-κB signaling pathway using the WB technique (Figure 8A). These pathways have been previously described to play an important role in bone formation [25,26].

Operation of the canonical Wnt/β-Catenin pathway causes the accumulation of β-Catenin in the cytoplasm. In the present study, the level of β-Actin-normalized β-Catenin protein was increased only in cases of SCAP co-cultured with *F. nucleatum* supernatant (*p* = 0.1705 ^ns^), although it was statistically significantly decreased in the case of SCAP co-cultured with viable *E. faecalis* (*p* = 0.0068 *) and non-statistically significantly decreased after co-culturing with viable *F. nucleatum* (*p* = 0.9258 ^ns^) (Figure 8B). Variation of the β-Catenin level in different treatment conditions was estimated using two-way ANOVA, with 60.496% (*p* = 0.4780 ^ns^) contributed by the treatment and 8.5140% (*p* = 0.0703 ^ns^) by the SCAP donor.

LRP-5 and LRP-6 are cell surface receptors of the Wnt/β-Catenin signaling pathway. Variation in the level of LRP-5 and LRP-6 expression could serve as a confirmation that this pathway is affected. The variation of the LRP-5 level in different SCAP treatment conditions revealed that the variation was 85.02% (*p* = 0.001 ***) and was caused by treatment. The donor of the SCAP did not play a significant role in the variation (7.9%; *p* = 0.1052 ^ns^). The levels of β-Actin-normalized LRP-5 protein in different variants of treatment were compared with the non-treated control using an unpaired t-test (Figure 8C). The LRP-5 level was statistically significantly decreased in cases of SCAP co-cultured with viable *F. nucleatum* (*p* = 0.0253 *) or viable *E. faecalis* (*p* = 0.0059 **). On the contrary, the LRP-5 level increased to the border of being statistically significant (*p* = 0.0942 ^ns^) in cases where SCAP were co-cultured with *F. nucleatum* supernatant.

The variation of β-Actin-normalized LRP-6 levels in different SCAP treatment conditions was similar to the variation of the LRP-5 levels. In cases of co-culture with viable bacteria, it decreased in SCAP co-cultured with *F. nucleatum* (*p* = 0.4148 ^ns^), as well as decreasing with *E. faecalis* (*p* = 0.0329 *), but it increased in cases of SCAP co-cultured with *F. nucleatum* supernatant (*p* = 0.1568 ^ns^) (Figure 8D). Two-way ANOVA analysis revealed that the variations of LRP-6 levels were 24.71% (*p* = 0.0747 ^ns^) caused by the SCAP donor and 57.32% (*p* = 0.0270 *) by the treatment.

pSMAD2/3 is a key mediator in the TGF- β signaling pathway. In this study, the level of pSMAD2 did not vary statistically significantly in different versions of co-culturing that used SCAP from the control level. In cases of SCAP co-cultured with viable *F. nucleatum*, the *p* was 0.2143; with viable *E. faecalis*, the *p* was 0.2815; and with the supernatant of *F. nucleatum*, the *p* was 0.9240 (Figure 8E). The variation in pSMAD2 levels in different treatment conditions was contributed to with 20.20% (*p* = 0.6788 ^ns^) from the treatment and 3.383% (*p* = 0.8781 ^ns^) from the SCAP donor.

NF-kB p65 expression by SCAP was decreased in all the variants of co-culturing in comparison with the non-treated control. The drop in NF-kB p65 expression was statistically significant in cases of SCAP co-cultured with *E. faecalis* bacteria (*p* = 0.0034 **) or with *F. nucleatum* supernatant (*p* = 0.0151 *). In cases of SCAP co-cultured with viable *F. nucleatum*, the level of NF-kB p65 expression did not fall statistically significantly (*p* = 0.1365 ^ns^) (Figure 8F). The variation of the NF-kB levels in different SCAP treatment conditions was statistically significant and, above all, caused by the treatment 59.50% (*p* = 0.0394 ^ns^), with only 18.22% (*p* = 0.1665 ^ns^) from the SCAP donor.

TNF receptor-associated factor 6 (TRAF6) plays an important role in the activation of certain intracellular signaling pathways upon exposure to bacteria to regulate immune responses via NF-kB, as well as in bone formation processes [27]. In the cases of SCAP co-cultured with viable bacteria, the TRAF6 expression was decreased: with *F. nucleatum* the decrease was statistically non-significant (*p* = 0.1778 ^ns^), and with *E. faecalis* it was statistically significant (*p* = 0.0285 *). The TRAF6 expression in SCAP cultured with the *F. nucleatum* supernatant increased slightly but not statistically significantly (*p* = 0.7135 ^ns^) (Figure 8G). Variation of the TRAF6 levels in different treatment conditions was 58.00% contributed to by the treatment, with a *p*-level on the border of being significant (*p* = 0.0583 ^ns^), and 15.65% (*p* = 0.2470 ^ns^) by the SCAP. 

## 3. Discussion

Stem cells interact with their niche, establishing a dynamic system that determines cell and tissue fate [28]. In the root canal, the presence of bacteria, as well as their residuals and toxins, is likely to impact the biology of SCAP and interfere with the success of regenerative endodontic treatments. In this context, we aimed to evaluate SCAP cytokine secretion profiles at the mRNA and protein levels in response to clinically relevant bacterial species, namely *F. nucleatum* and *E. faecalis*. Our results suggested that bacteria associated with root canal infection modulates SCAP cytokine profiling in a species-dependent manner. Accordingly, *F. nucleatum* triggered, while *E. faecalis* inhibited, pro-inflammatory chemokine and cytokine generation, such as IL-6, IL-8, and MCP-1 at the mRNA and protein levels. Further, *F. nucleatum* upregulated immune and inflammatory responses, and *E. faecalis* downregulated the levels of key proteins of the Wnt/β-Catenin signaling pathway, which plays a crucial role in bone formation. 

In agreement with our previous findings [22] using a V-PLEX human biomarker multiplex assay, *F. nucleatum*, but not *E. faecalis*, induced cytokine production of the pro-inflammatory chemokines IL-8 and IL-10 in SCAP. In the current study, we showed that viable *F. nucleatum* upregulated the secretion of chemokines and cytokines such as GM-CSF, GROα, IL-6, IL-8, MCP-1, and VCAM-1, as detected by microarray and transcriptomics. Our results are consistent with previous studies using different cells lineages. Wang and Zhao [29] reported that *F. nucleatum* triggered an inflammatory response in human leukemic monocyte cells by upregulating the expression of TNF-α, IL-1β, IL-6, and IL-8. Moreover, *F. nucleatum* infection led to the recruitment of macrophages and osteoclasts, resulting in gingival inflammation and bone resorption [30]. In human gingival epithelial cells (GECs), *F. nucleatum* promoted chemokine and cytokine generation, such as CCL2, CCL20, CXCL1, PTGS2, IL-6, IL-8, and IL-1β, at the mRNA level [31]. However, further studies exploring the detailed molecular mechanism by which *F. nucleatum* promotes inflammatory cytokine production in SCAP are warranted.

Evidence indicated that MSCs may affect neighboring innate and adaptive immune cells in two main ways: direct cell–cell contact and the release of a variety of soluble factors. We found that viable bacteria (direct contact) and supernatants (soluble factors) of *F. nucleatum* in SCAP did not vary significantly. However, Dkk-1 and SDF-1α were secreted by SCAP in indirect contact with *F. nucleatum*. Dkk-1 rescues the osteogenic differentiation of mesenchymal stem cells [32], suggesting that *F. nucleatum* metabolites have a negative influence in the regenerative endodontic treatment.

*F. nucleatum* is considered an important microorganism for maintaining the integrity of a healthy oral microbiome but is increasingly being implicated as one of the key species in the development of periodontitis and primary root canal infection [14,33]. In cases of dysbiosis in oral microbiota, the progressive development of periodontitis can occur [34]. Interestingly, the use of probiotics as an alternative to chlorhexidine in adjuvant therapy for standard periodontitis treatment can reduce bacterial counts of *F. nucleatum* in long-term use (up to six months) [35]. Accordingly, using toothpaste and chewing gum that contain probiotics of the genus *Lactobacillus* and *Bifidobacterium* as a support for the gold standard treatment led to a significant decline in the orange complex pathogens of *F. nucleatum* and *Prevotella intermedia* [35]. Since highly regulated signals are modulated by a network of microbial and host metabolites, new strategies adding the proactive action of probiotics are desirable to balance the oral microbiota and maintain eubiosis constancy.

When SCAP were co-cultivated with viable *E. faecalis*, neither IL-8 nor IL-6 was detected. Notably, SCAP did not express cytokine mRNA and translate it into protein when cells were co-cultured with *E. faecalis.* Such a situation, when mRNA is present but protein is not detected, could be explained by different post-translational modifications or limited detection levels of proteins [36,37]. For the majority of the secreted and detected cytokines, mRNA was not detected, suggesting regulation of these cytokines’ secretion by negative feedback loops [38]. At the protein level, *E. faecalis* upregulated the production of only three cytokines: Emmprin, MIF, and endoglin, in SCAP. Emmprin is an extracellular matrix metalloproteinase inducer which facilitates the secretion of MMP-1, MMP-3, MMP-9, and membrane-type 1-MMP [39]. Emmprin plays an important role in inducing protection against bacteria via the MAPK signaling pathway [40]. In vitro, MIF induced monocyte-specific TNF production and enhanced mineralization and osteoblastic gene expression in osteoblast cell lines [41,42]. Endoglin is part of the TGF-β pathway [43], which has an important function to control immune response. This pathway leads to a decrease in the immune response [44]. The pathway is activated by a TGF- β cytokine that was not included in the set of 105 cytokines applied on the membrane, but its secretion was analyzed in an earlier study [22].

To date, the interactions of *E. faecalis* with SCAP and the effects of such interactions on the body’s immune response and bone formation are not clear. In our study, we confirmed that *E. faecalis* significantly inhibited SCAP inflammatory and immunomodulatory response and decreased SCAP osteogenic differentiation, partly by downregulating osteogenesis-related gene and protein expression and by decreasing key protein-related bone formation signaling pathways. The comparative GO analysis revealed that, in SCAP co-cultured with *E. faecalis,* the processes of cell adhesion were specifically downregulated, as was cellular response to hypoxia. Cytokine production is a crucial component for host defense against pathogenic invasion [45]. The ability of *E. faecalis* to downregulate the immune and inflammatory responses indicated a high virulence of the strain [46]. Consistent with previous studies, *E. faecalis* influenced bone marrow stem cell differentiation by promoting CD11c-positive dendritic cells with aberrant immune functions retaining the capability of proinflammatory cytokine induction [47]. Another study showed that *E. faecalis* in high doses slowed healing and suppressed the expression of inflammatory cytokines in a mouse model [48]. *E. faecalis* altered the host immune response to facilitate infection with other pathogens and caused immune suppression of macrophages [49]. Unlike this study, Molina and Díaz [50] demonstrated that the immune stimulation triggered by *E*. *faecalis* CECT7121 involved the activation of the innate response in dendritic cells and development of Th1-related cellular adaptive responses. They indicated that even different strains that belong to the same species could modulate cellular functions distinctly. 

We further investigated key proteins of SCAP involved in signaling pathways, such as the Wnt/β-Catenin, TGF-β, and NF-κB signalling pathways, relevant in bone formation. It is known that the Wnt/β-Catenin pathway is implicated in the pathogenesis of several diseases, including impaired bone healing, autoimmune diseases, and malignant degeneration [32]. However, it is still unclear what role Wnt signalling has in root canal infection and tooth regeneration. The Wnt/β-Catenin pathway specifically regulates stem cell differentiation during the initiation of many tissues and organs, including the regulation of dental stem cells in tooth development [51,52]. Wnt co-receptor LRP5-mediated Wnt/B-Catenin signaling is known to be involved in bone metabolism and bone growth through its effects on the increased differentiation of mesenchymal stem cells into osteoblasts and preventing apoptosis of osteoblasts [53]. Loss of function missense mutation in LRP5 causes severe osteoporosis, a condition called osteoporosis-pseudoglioma syndrome (OPPG) [54]. In the present study, statistically significantly decreased levels of LRP5, LRP6, and β-Catenin in SCAP upon direct contact with *E. faecalis* were observed, leading to the downregulation of key signaling components in the Wnt/B-Catenin pathway. These results, along with previous studies, showed that decreased Wnt/B-Catenin, through downregulation of its receptor, LRP5, might affect the differentiation and maturation of SCAP cells in the presence of *E. faecalis*. Similar results were also observed in NF-kB signalling components such as TRAF6 and NF-kB p65, in which TRAF6, through its effect on RANKL-mediated NF-kB activation, had a significant role in bone remodeling and resorption [27,55]. Wnt/β-Catenin is considered an important target of several virulence factors produced by bacteria [56]. The structural details of virulence factors need to be elucidated in order to design specific drugs to deactivate these virulence factors and enhance regeneration treatment.

The cytokine induction profile of stem cells has been previously described to be dependent on the cell type, bacterial species, and methodology used (e.g., period of stimulation) [6]. However, as we used an in vitro system, we were aware that we would not be able to mimic the situation of chronic inflammation in a tissue where there were several inflammatory cells acting in concert with each other, as well as the surrounding tissue and the cells. Thus, the extrapolation of results to mimic the situation of a chronic process should be performed with care. There are limitations of an in vitro study, such as a not necessarily accurate reflection of physiological process and taking into consideration the complex biofilm in the root canal; in the future, the investigation of a large number of species, establishment in vivo models, and an analysis of the persistence of infection would be warranted. Our approach can still provide useful information to enhance understanding of in vivo processes.

One of the strengths of this study was the fact that we investigated the biological properties and mechanism of SCAP under hypoxia and infection conditions, simulating the in vivo environment. Indeed, if dental pulp becomes infected, it leads to a hypoxic environment in the root canal, which may impair MSC function and weaken osteo-odontogenic differentiation [57]. Interestingly, some studies have shown decreased osteoblast formation and mineralization under hypoxic condition [58,59], and other studies have demonstrated that hypoxia conditions could upregulate osteogenic-related genes in MSCs derived from dental tissues [60,61]. 

We confirmed that oral bacteria isolated from root canals played crucial roles in altering the cytokine profile and functional properties of SCAP in a species-dependent manner. Thus, *F. nucleatum* triggered, while *E. faecalis* inhibited, chemokine and cytokine generation, such as IL-6, IL-8, and MCP-1, at the mRNA and protein levels. Moreover, we discovered that *E. faecalis*-infected SCAP significantly downregulated and *F. nucleatum* upregulated immune response, inflammatory response, and response to hypoxia processes, which could suggest a proactive effect. Additionally, the regulation of protein expression for key components in the NF-kB and Wnt pathways played a critical role in adult osteogenic differentiation of mesenchymal stem cells, as they were significantly downregulated in *E. faecalis*-infected SCAP, suggesting a high virulence of the strain and the success of RET (regenerative endodontic therapy).

These findings highlighted the negative impact that infection and hypoxia can have on SCAP in the success of regenerative endodontic therapy. Our study illuminated the biological effects of *F. nucleatum* and *E. faecalis* on SCAP and established a basis for future mechanistic studies of SCAP under oral bacterial infection. 

## 4. Materials and Methods

### 4.1. Bacterial Strains, Supernatants, and Growth Conditions

Clinical isolates of *F. nucleatum* subsp. *polymorphum* and *Enterococcus faecalis* were used in this study for their impact on SCAP [22]. 

The bacterial isolates were obtained as previously described [13]. Briefly, the contents of the root canal were absorbed into sterile paper points and transferred to fluid thioglycolate medium supplemented with agar. The paper points were moved to TE buffer and cultured on fastidious anaerobic agar (FAA, Svenska LABFAB, #ACU-7531A) supplemented with 5% citrated bovine blood and 1% vitamin K in an anaerobic atmosphere (5% CO_2_, 10% H_2_, and 85% N_2_ at +37 °C) for one week. The strains were identified by comparing the 16S rRNA gene sequence to databases (HOMD) and by MALDI-TOF MS analysis using a Voyager DE-STR MALDI-TOF instrument (AB Sciex, Umeå University) with sinapinic acid as the matrix.

The bacteria used in this study were taken from cryostocks (−80 °C) and streaked at least three times on FAA prior to the experiments. Freshly grown bacteria colonies were resuspended in an antibiotic-free cell medium (α-MEM, GlutaMAX^TM^) and supplemented with 10% fetal bovine serum (FBS; GIBCO). The optical density of each suspension was adjusted to OD_600_ = 1.0 (1 × 10^8^ cells/mL). 

To obtain bacteria cell-free supernatant, fresh bacterial suspensions (OD600 = 1.0) were inoculated on antibiotic-free cell culture until desirable MOI were reached. After 24 h of incubation (5% CO_2_, 10% H_2_, and 85% N_2_ at 37 °C), the supernatants were collected by centrifugation (10,000× *g* during 10 min at 4 °C), filter-sterilized through a syringe membrane filter (pore size 22 µm), aliquoted, and stored at −80 °C until use.

### 4.2. Isolation, Culture, and Characterization of Human SCAP

The SCAP were isolated from three healthy human donors referred to as SCAP donor I, SCAP donor II, and SCAP donor III, as previously described [22,62]. In brief, human impacted teeth were surgically removed from healthy patients due to retention or lack of space in orthodontic treatment. The teeth and surrounding tissues were placed in a tube with minimum essential medium alpha modification (α-MEM) and GlutaMAX™ (GIBCO/Invitrogen, Carlsbad, CA, USA), supplemented with 1% antibiotic-antimycotic solution (Sigma-Aldrich, St. Louis, MO, USA), and transferred to a laboratory within 4 h. To collect the SCAP, the apical papilla was gently removed from the teeth with a scalpel, minced into small pieces, and treated by dissolving all the tissues in a solution of dispase II and collagenase I, followed by filtration. The flow-through was diluted and transferred to a 25 cm^2^ tissue culture plastic flask (Thermo-Fisher Scientific, Hvidovre, Denmark) and incubated at 37 °C with 5% CO_2_. The cells were then detached and transferred to 75 cm^2^ flasks (Thermo Scientific) and allowed to grow until 90% confluency, with changing of the medium every other day.

Cells at passage 2 that were cryopreserved for three years were used in this study, and all comparisons were made between cells at matching passage numbers. The authenticity of the multipotent stromal cells was confirmed by the presence of CD73, CD90, CD105, and CD146 and the absence of CD11b, CD19, CD34, CD45, and HLA-DR. The collection, culture, storage, and usage of all the cell lines were approved by the local research ethics committee at Umeå University (Reg. no. 2013-276-31M). The absence of *Mycoplasma* species in the studied SCAP was confirmed using a Venor GeM Mycoplasma Detection Kit that was PCR-based (Sigma-Aldrich, St. Louis, MO, USA, MP0025). 

Directly derived from cryostock, the cells were cultured on α-MEM GlutaMAX™ medium supplemented with 10% FBS and 1% penicillin-streptomycin solution (Sigma-Aldrich, St. Louis, MO, USA) at 37 °C under 5% CO_2_ atmospheric conditions until 95% confluency. The SCAP were then detached by trypsin, collected, and counted with a Countess™ II Automated Cell Counter. The cells (40,000 live cells/mL) were seeded and incubated overnight at +37 °C under 5% CO_2_ atmospheric conditions. On the next day, the adherence of the SCAP was verified under a microscope, followed by an exchange of the antibiotic-free culture media.

### 4.3. Variant of Treatments Included in Co-Culture Experiments

Variants of the treatment for profiling of the cytokines were chosen, adjusting the multiplicity of infection (MOI) to reflect the clinical situation with regard to adverse effects, as previously described [22]. Accordingly, the cytotoxic effect of viable bacteria at the planktonic stage was evaluated on SCAP using a neutral red dye assay with a wide range of bacterial concentrations (MOIs of 0.1, 1, 5, 10, 50, and 100). Moreover, a concentration sufficient to inhibit cell proliferation by 50% (IC50) after 24 h of SCAP–bacteria co-culture was calculated using a xCELLigence Real Time Cell Analyser (RTCA, Roche Diagnostics GmbH, Mannheim, Germany). Taken together, only SCAP exposed to viable *F. nucleatum* and *E. faecalis*, as well as *F. nucleatum* supernatants, at concentrations of MOI 100 expressed visible effects (*p* < 0.05) and were chosen for this study (Appendix A). 

Samples from the Donors (I-III) were pooled to reduce individual variances [63], and four variants of treatment were analyzed for cytokine profiling: (i) SCAP co-cultured with viable *F. nucleatum*; (ii) SCAP co-cultured with viable *E. faecalis*; (iii) SCAP co-cultured with *F. nucleatum* supernatant, and (iv) SCAP without bacteria (used as controls).

For the co-culture experiments, viable bacteria or their supernatants were resuspended in antibiotic-free cell culture medium and adjusted to MOI 100 in SCAP, as previously described [22]. After 24 h of co-culture at +37 °C under anaerobic conditions, the conditioned medium was collected by centrifugation (800× *g* for 10 min at room temperature), aliquoted, and stored at −20 °C until investigation. 

### 4.4. Detection of Secreted Cytokines by Protein Array

To identify cytokines and chemokines expressed in SCAP in response to viable bacteria or supernatants, a Human XL Cytokine Array Kit (RnD system, ARY022B) was used with 105 captured antibodies (listed in the Appendix A) that were spotted in duplicate on a nitro-cellulose membrane, according to the manufacturer’s protocol. Then, immunospots were captured with an Amersham™ Imager 600 (GE Healthcare, Chicago, IL, USA), and the raw data were analyzed using Excel. To calculate the relative pixel intensity for each cytokine, the following formula was applied:$$\bar{x}=\frac{x_{1}-x_{negative\;control}}{x_{1\;positive\;control}}$$

### 4.5. mRNA Preparation and Transcriptome Sequencing Analysis

The total RNA was extracted using an RNA extraction kit (RNeasy Plus Mini Kit Qiagen, 74134) according to the manufacturer’s instructions. RNA purity and concentration were measured using RNA screentape from a TapeStation system (Agilent Technologies, Santa Clara, CA, USA). The preparation of the RNA library and transcriptome sequencing were conducted by Novogene Co., Ltd. (Beijing, China). 

The raw sequences were trimmed for quality, and the adapters were cut using cutadapt [64]. Then, the good quality sequences were mapped to the human genome (GRCh38.p13) using HISAT2 [65]. RSeQC [66] was used to check for quality control of the mapping statistics, and then the counts of the exons were calculated using featureCounts [67]. The genes with very low counts (i.e, genes with counts <5 in <3 samples) were removed from the data. As the variations between the Donors (I–III) were a relatively dominant factor in the expression of the SCAP, the variations due to the Donors were removed using the “removeBatchEffect” function from limma [68] before they were transformed with VST (variance stabilizing transformation) and visualized using PCA. The differential expression analysis was performed using DESeq2 [69] on the filtered counts with the design matrix (~Donor + Treatment) where “Treatment” was a specific co-culture experiment.

### 4.6. Protein Interaction Mapping

To model molecular mechanisms of the biological processes, protein–protein interaction maps were created using a free web resource, Proteomaps (https://www.proteomaps.net/index.html, accessed on 24 May 2021) [24]. Proteomaps shows the quantitative composition of proteomes with a focus on protein function. The upregulated and downregulated molecular mechanisms detected by Proteomaps were compiled into one diagram to visualize the perturbation of molecular mechanisms in SCAP upon oral bacteria exposure. Additionally, polygon areas represented protein abundances and were weighted by protein size.

The Database for Annotation, Visualization and Integrated Discovery (DAVID; version 6.8; david.ncifcrf.gov/, accessed on 27 May 2021) [70] was used to perform gene ontology (GO) enrichment (www.geneontology.org, accessed on 27 May 2021). 

### 4.7. Preparation of Total Cell Lysate for SDS-PAGE Electrophoresis

After 24 h of co-culturing SCAP with viable bacteria or their supernatant, the cells were rinsed 3 times with sterile, ice-cold, phosphate-buffered saline (PBS) and lysed in ice-cold lysis buffer (0.15 M NaCl, 1% NP-40 alternative, 1% sodium oxycholate, and TRIS 25 mM (pH 7.4) with 1 mM aprotinin, 1 mM Pefabloc, and 1 mM sodium orthovanadate). The cell lysates were aliquoted and stored at −80 °C until further investigation. LDS sample buffer was used for total cell lysate sample preparation.

### 4.8. SDS-PAGE and Immunoblot Analysis

The protein concentrations in the total cell lysates were determined using a Pierce BCA Protein Assay kit (Thermo Fisher Scientific, Waltham, MA, USA) in order to load equal amounts of total protein per lane in NuPAGE 7% Tris-Acetate polyacrylamide precast gels (Invitrogen) using Tris–Acetate as a running buffer (Invitrogen). After size fractionation in denaturing and reducing conditions, the proteins were electrophoretically transferred to a nitro-cellulose membrane (Invitrogen) using an iBlot 2 Dry Blotting System (Invitrogen), and the blots were incubated in Odyssey^®^ blocking buffer (TBS) for 1 h. Specific proteins were detected using primary antibodies. The primary antibodies were detected using either IRDye^®^ 800CW goat anti-rabbit or IRDye^®^ 680RD goat anti-mouse (1:5000) from Licor Biosciences and were visualized using a LiCor Odyssey infrared Imaging System.

The protein densities were measured using Image Studio software. The ratio between the density of the protein of interest and the density of β-Actin as the constitutively expressed protein was used to calculate the difference between samples. Quantification was based on three or more independent experiments.

### 4.9. Antibodies and Reagents

The following antibodies and reagents were used in the experiments:

Anti LRP5 (D80F2) rabbit monoclonal antibody (#5731) and Monoclonal Anti-LRP6 (C5C7) rabbit antibody (#2560) were purchased from Cell Signaling technology (Denvers, MA, USA). Anti-β-Catenin monoclonal mouse antibody (#610154) was obtained from BD Bio-sciences (San Jose, CA, USA), and Mouse monoclonal anti-β-actin (#A5441) was obtained from Sigma-Aldrich (St. Louis, MO, USA). Anti-Rabbit Polyclonal NF-kB p65 antibody (ab16502) and Monoclonal rabbit anti-TRAF6 (#ab40675) antibody were obtained from Abcam (Cambridge, MA, USA). Monoclonal anti-Rabbit Phospho-SMAD2 (Ser465/467) (138D4) (#3108) was purchased from Cell Signalling technology (Denvers, MA, USA). The secondary Antibodies IRDye^®^ 800CW goat anti-Rabbit (#925-32211) and IRDye^®^ 680RD goat anti-Mouse (#925-68070) were obtained from Licor Biosciences (Lincoln, NE, USA).

Pefabloc (#11429876001)) was purchased from Roche (Basel, Switzerland) and sodium orthovanadate from Sigma-Aldrich. Spectra Multicolor High Range Protein ladders were purchased from Thermo Fisher Scientific (Waltham, MA, USA). LDS sample buffer was obtained from Invitrogen., 4, 6-Diamidino-2-phenylindole dihydrochloride (DAPI) was obtained from Vector Laboratories.

### 4.10. Statistical Analysis

GraphPad software (GraphPad Prism for Windows, version 8.4.3) was used for the statistical analysis of the calculated values. The data are presented as mean values ± standard deviation (SD) or as median values ± confidential interval (CI); *p* ≤ 0.05 was considered to be statistically significant. To compare variables, an unpaired t-test or two-way ANOVA was used. 

## Figures and Tables

**Figure 1 ijms-23-05098-f001:**
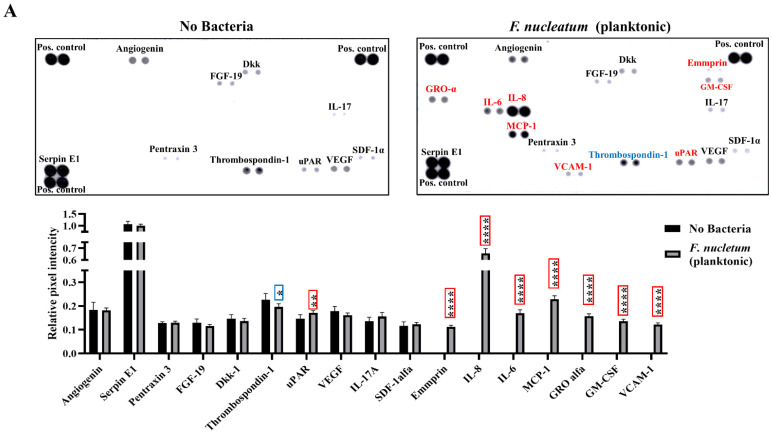
Cytokine and chemokine profiling of SCAP upon bacterial exposure. Images of membranes treated with SCAP infected by (**A**) *F. nucleatum* (planktonic), (**B**) *F. nucleatum* supernatants, or (**C**) viable *E. faecalis* (planktonic). Red text represents upregulation in comparison with the non-treated control. Blue represents downregulation of cytokines and chemokines. Difference between the relative intensity of immunospots of corresponding cytokine and chemokine in different SCAP culture variants is illustrated in staples: * (*p* < 0.01), ** (*p* < 0.001) or **** (*p* < 0.00001). Data are presented as the mean  ±  SD.

**Figure 2 ijms-23-05098-f002:**
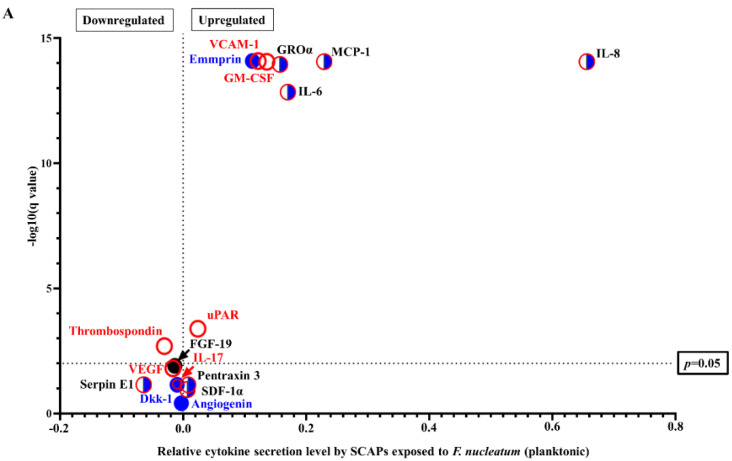
Immunomodulatory- and osteogenesis-related chemokine and cytokine levels by SCAP modulated by oral species. Volcano plots representing SCAP modulated by viable (**A**) *F. nucleatum* (planktonic), (**B**) *F. nucleatum* supernatants, or (**C**) viable *E. faecalis* (planktonic). Osteogenesis-related cytokines are marked by circles filled with blue; immunomodulatory-related cytokines are marked by empty circles with red borders; cytokines that can play a role in osteogenesis and in immunomodulation are marked by half-blue and half-white circles with red borders; cytokines with a function related to other processes are marked by circles filled with black.

**Figure 3 ijms-23-05098-f003:**
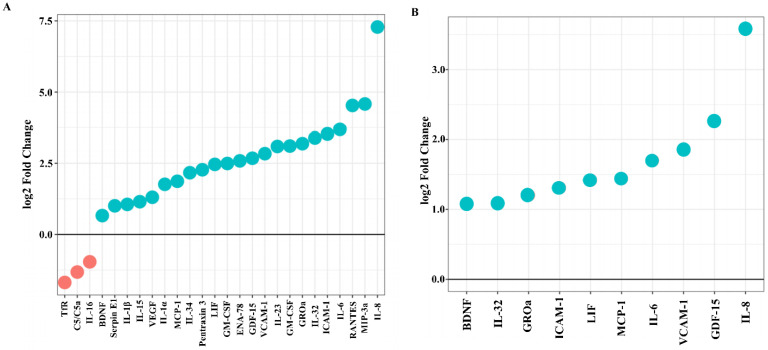
mRNA expression of cytokines and chemokines by SCAP modulated by oral bacteria. SCAP co-cultured in direct contact with (**A**) *F. nucleatum*, (**B**) *F. nucleatum* supernatants, and (**C**) *E. faecalis* with corresponding untreated control levels. Blue and red dots indicate upregulated and downregulated cytokines, respectively.

**Figure 4 ijms-23-05098-f004:**
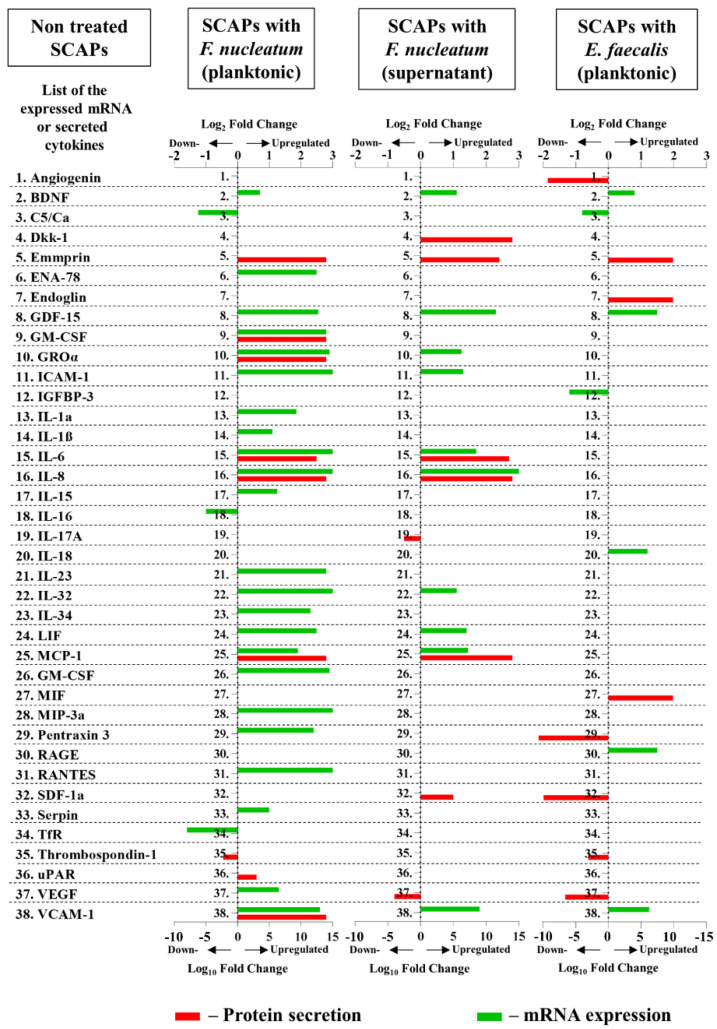
Up- and downregulation of the cytokines by SCAP with different variants of treatment detected at mRNA and protein levels. The regulation of protein secretion is expressed in log_10_ fold change, and the corresponding mRNA expression is expressed in log_2_ fold change.

**Figure 5 ijms-23-05098-f005:**
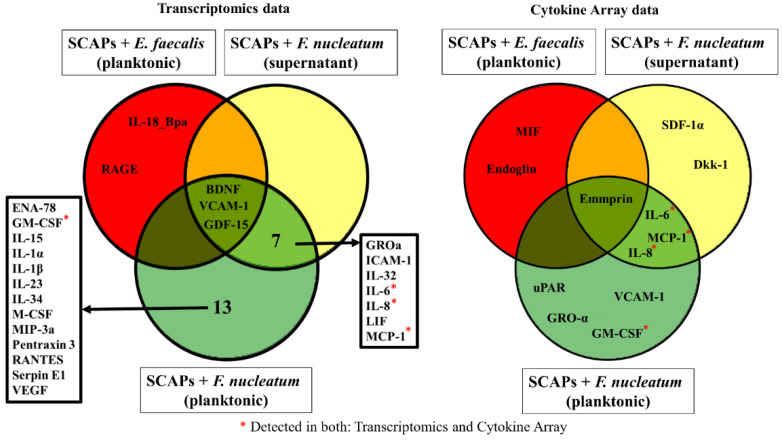
Venn diagrams showing upregulated chemokines and cytokines commonly expressed at mRNA and protein levels. The diagrams show the cytokine secretion at mRNA (on the left) and protein levels (on the right) by SCAP modulated by oral bacteria with different variants of treatment compared with corresponding control levels.

**Figure 6 ijms-23-05098-f006:**
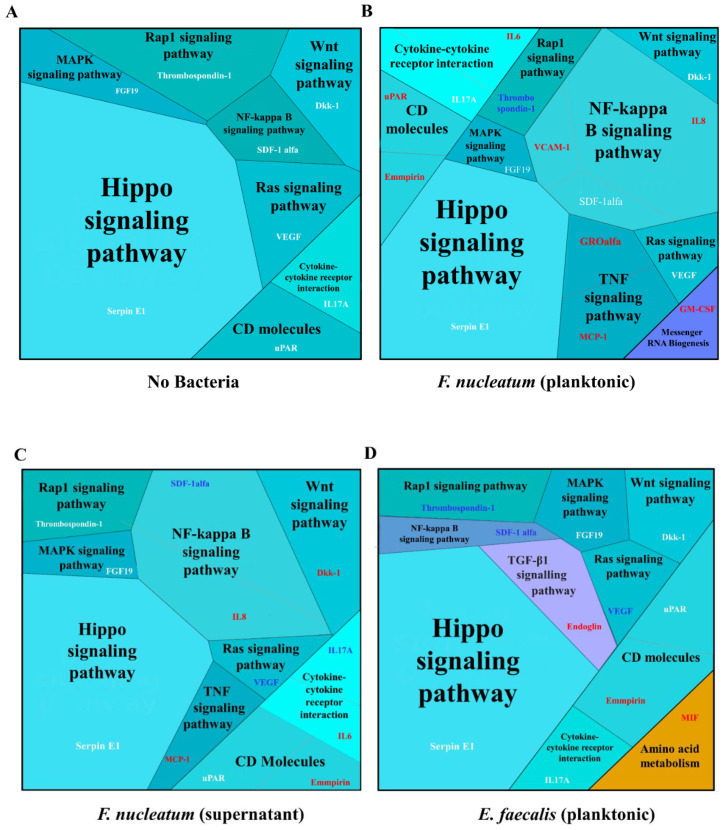
Proteomap predicted different signaling pathways and molecule interactions associated with SCAP modulated by oral bacteria. (**A**) No bacteria, (**B**) *F. nucleatum (planktonic)*, (**C**) *F. nucleatum* supernatants, and (**D**) *E. faecalis* (planktonic) with corresponding untreated control levels. Upregulated proteins are marked by the red text; downregulated cytokines are marked by blue text; cytokines that were secreted at a similar level as the non-treated control are marked by white text. Each protein is shown by a polygon, and functionally related proteins are arranged in common regions. To emphasize highly expressed proteins, polygon areas represent protein abundances weighted by protein size. In each Proteomap, angiogenin and pentraxin 3 are omitted.

**Figure 7 ijms-23-05098-f007:**
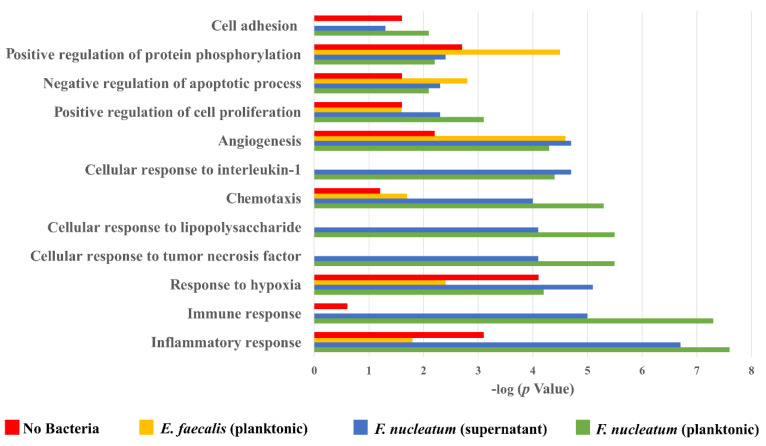
GO annotated biological processes in different variants of SCAP treatment. Coverage in each variant is 100%.

**Figure 8 ijms-23-05098-f008:**
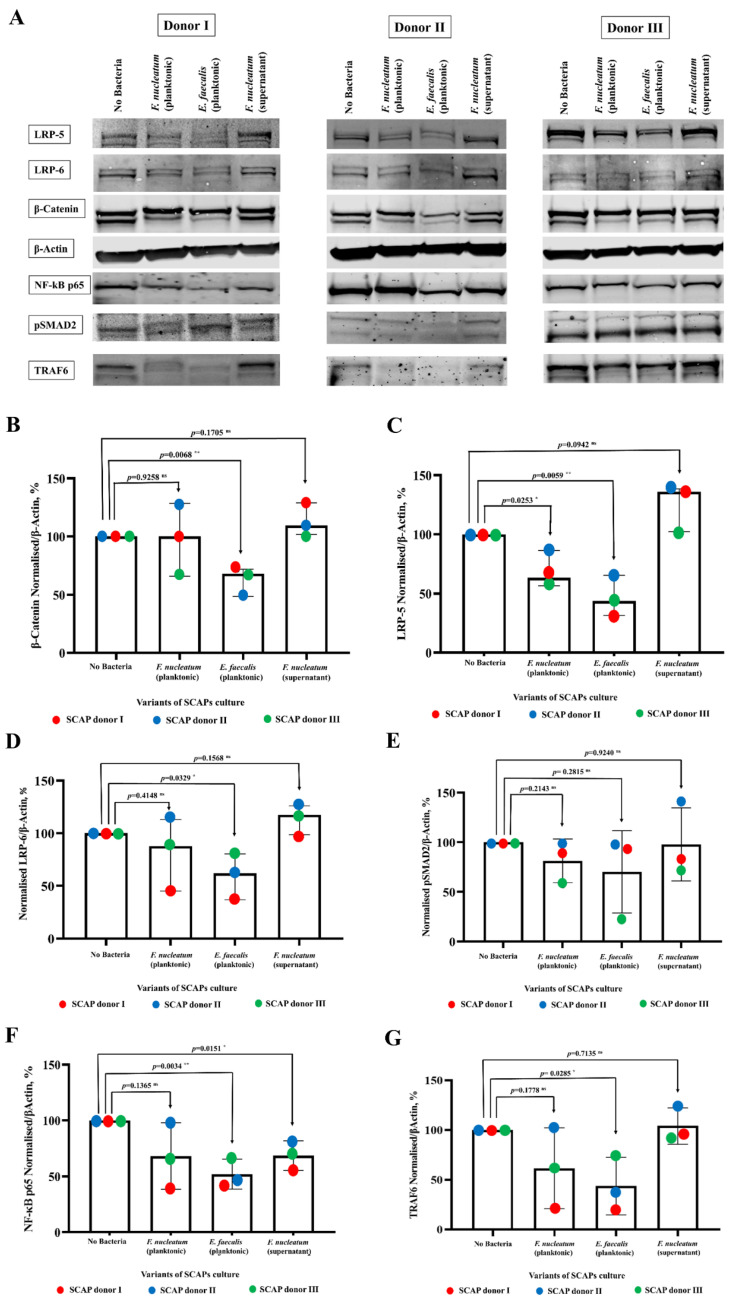
Alterations in expression of LRP-5, LRP-6, β-Catenin, pSMAD2, TRAF6, and NF-κB p65 in SCAP modulated by *F. nucleatum* (planktonic and supernatants) or *E. faecalis* (planktonic). (**A**) Cells were cultured in designated conditions, collected, and lysed. Expression of listed proteins was estimated by the Western blot technique. Relative expression of proteins for (**B**) β-Catenin, (**C**) LRP-5, (**D**) LRP-6, (**E**) pSMAD2, (**F**) NF-κB p65, and (**G**) TRAF6 in SCAP from donors I–III upon co-culture in designated conditions (amount of proteins in each co-culture condition is expressed as a percentage, and amount of protein in the absence of bacteria was set as 100%). Values represent the medians with confidence interval (CI) and were compared using unpaired t-test. Differences between variables are marked by the corresponding *p* values with symbols: * *p* < 0.01 or ** *p* < 0.001 in the case of statistically significant or ^ns^*p* > 0.05 in the case of statistically non-significant.

## Data Availability

The raw data supporting the conclusions of this article will be made available by the authors without undue reservation.

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
