# Peer review of "Combined Transcriptomic and Protein Array Cytokine Profiling of Human Stem Cells from Dental Apical Papilla Modulated by Oral Bacteria"

_ijms, 2022, doi:10.3390/ijms23095098_

Round 1

Reviewer 1 Report

Manuscript of considerable interest for the entire dental sector.

Changes to be made before being submitted to publications

Add specific keywords

Introduction: to argue the pathogenic factor of F. Nucleatum and to those bacterial strain belongs, if we are talking about periodontitis, which stages and degrees? Based on the new classification of periodontal disease

Results: very dispersive, and tables not easy to interpret by the common reader, to reorganize them and make them more visible in the data to be emphasized.

Grainy graphics and not easy to understand, to add the originals and not the files in jpeg.

Well described materials and methods.

Discussion, to add the proactive action to reduce the incidence of F. nucleatum through the supply of probiotics, paraprobiotics and postbiotics to balance the oral microbiota and maintain an eubiosi constancy. the research group of Scribante et al. they dedicated themselves to these studies

Conclusions: proactive action to be added

Bibliography: references reported in the discussion to be added

Author Response

Thanks for your comments. Please find in the attached documment the cover letter providing point-by-point response to the reviewer´s comments.

Reviewer 2 Report

General comment: The authors covered a current and interesting topic in the field of endodontics and organized the paper comprehensively! This manuscript has the merit of being published in the IJMS. Only a few things to correct; in line 67 put before the full name of IL-6, IL-8 and delete the long names of them in line 112.

Author Response

(The authors gave the same response as above.)

Round 2

Reviewer 1 Report

The authors of the manuscript have reported the required changes, the only incompleteness perhaps is the lack of citation on para Probiotics which, if desired, can always be added after citation 35, always from the same research group